# Angiogenesis and Functional Vessel Formation Induced by Interstitial Flow and Vascular Endothelial Growth Factor Using a Microfluidic Chip

**DOI:** 10.3390/mi13020225

**Published:** 2022-01-29

**Authors:** Yufang Liu, Jiao Li, Jiasheng Zhou, Xue Liu, Huibing Li, Yao Lu, Bingcheng Lin, Xiaojie Li, Tingjiao Liu

**Affiliations:** 1Department of Oral Pathology, Shanghai Stomatological Hospital & School of Stomatology, Fudan University, Tianjin Road No. 2, Huangpu District, Shanghai 200001, China; 18241652728@163.com (Y.L.); jiaoli@fudan.edu.cn (J.L.); liuxue@fudan.edu.cn (X.L.); 2Shanghai Key Laboratory of Craniomaxillofacial Development and Diseases, Fudan University, Tianjin Road No. 2, Huangpu District, Shanghai 200001, China; 3School of Stomatology, Dalian Medical University, West Section No. 9, Sourth Road of Lvshun, Dalian 116044, China; chowjason1993@dmu.edu.cn (J.Z.); lihuibing@dicp.ac.cn (H.L.); 4Department of Biotechnology, Dalian Institute of Chemical Physics, Chinese Academy of Sciences, Zhongshan Road No. 456, Dalian 116023, China; luyao@dicp.ac.cn (Y.L.); bclin@dicp.ac.cn (B.L.)

**Keywords:** angiogenesis, interstitial flow, VEGF, microfluidic

## Abstract

Angiogenesis occurs during both physiological and pathological processes. In this study, a microfluidic chip for the development of angiogenesis was utilized to assess angiogenic sprouting and functional vessel formation. We also found that vascular endothelial growth factor (VEGF) was a determinant of the initiation of vascular sprouts, while the direction of these sprouts was greatly influenced by interstitial flow. Isoforms of VEGF such as VEGF_121_, VEGF_165_, and VEGF_189_ displayed different angiogenic properties on the chip as assessed by sprout length and number, vessel perfusion, and connectivity. VEGF_165_ had the highest capacity to induce vascular sprouting among the three isoforms assessed and furthermore, also induced functional vessel formation. This chip could be used to analyze the effect of different angiogenic factors and drugs, as well as to explore the mechanism of angiogenesis induced by such factors.

## 1. Introduction

Angiogenesis refers to the process of vessel sprouting and growth from pre-existing blood vessels induced by a series of angiogenic factors [1,2]. Stimulated by proangiogenic signals, endothelial cells differentiate into motile and invasive tip cells that grow protruding filopodia. Tip cells are located at the front of new sprouts and guide their direction of growth. Following tip cells, stalk cells proliferate to support the elongation of sprouting and can establish a lumen. Tip cells from neighboring sprouts fuse with each other to build vessel loops (anastomosis), and within the lumen extension and blood flow initiation, the newly formed vessels become perfused [2,3,4,5]. Connectivity and perfusion are important indices with which to evaluate the function of blood vessels.

Vascular endothelial growth factor (VEGF) is one of the most important angiogenic factors [6]. The gene encoding human VEGF, VEGF-A, includes eight exons and can generate several isoforms containing 121~206 amino acids through alternative splicing of exons 6 and 7. In this way, several isoforms, including VEGF_121_, VEGF_165_, and VEGF_189_, are formed, and VEGF_165_ is the most frequently expressed in normal tissues [7]. These isoforms are important in both normal embryonic and adult development, as well as the progression of some diseases as they perform unique angiogenic functions [8,9,10]. However, the angiogenetic characteristics induced by the three different isoforms remains unclear.

Fluid flows within the interstitium of the extracellular matrix (ECM) and transports nutrients and signaling molecules between blood vessels, lymphatic capillaries, and the ECM [11]. Interstitial flow (IF) is thought to be approximately 0.1–1 μm/second in normal tissues but increases in various pathological conditions [12,13]. IF plays a critical role in both pathological and physiological angiogenesis and can generate gradients of angiogenic factors (e.g., VEGF), which can potentially stimulate and guide vessel sprouting [14]. In addition, IF affects the migration of cells by transducing the mechanical forces experienced by the cells [15].

The study of angiogenesis requires experimental platforms capable of mimicking the in vivo microenvironment as closely as possible. An ideal platform will enable researchers to study sprouting initiation, and importantly, to evaluate vessel function. Microfluidic chip platforms have unique advantages for the study of angiogenesis, as they combine 3D tissue scaffold structures, chemical gradients, fluid flow forces, and intercellular interactions, all of which can be integrated to provide a physiologically relevant context [16,17]. Researchers use microfluidic models to reconstitute the morphogenetic steps involved in angiogenic sprouting, including tip cell differentiation, stalk cell elongation, and lumen formation [18]. The establishment of a perfused lumen is an important index for the construction of functional blood vessels in vitro [19,20]. Perfusion and connectivity are important indices with which to assess the maturation of newly formed blood vessels [21,22]. Tube formation assays are the most used in vitro model to study angiogenesis. However, some key events during the angiogenic processes, such as guided sprouting, lumen formation, and perfusion, can hardly be achieved using this model [23]. Microfluidic chips have emerged as an important technology for biomimicry of the cellular microenvironment in vivo [24]. Functional vessel models constructed using the microfluidic technique can be employed for various functional tests such as the effect of blood flow on endothelial cells, cancer angiogenesis and intravasation, and the screening of anti-angiogenesis drugs [25,26,27,28,29]. Blood vessel sprouting and even lymphatic vessel sprouting have been reproduced on these microfluidic models [30,31]. However, it is still not simple to produce a functional vessel on these microfluidic chips.

Here, we demonstrated a method to reconstruct an angiogenic model using a microfluidic chip, on which both the morphology of angiogenic sprouting and vessel functions, such as connectivity and perfusion, could be studied. We demonstrated that IF was capable of not only enhancing the angiogenic capacity of VEGF but also guiding the direction of vascular sprouting. We further compared the vascular sprouts induced by human isoforms of VEGF (including VEGF_121_, VEGF_165_, and VEGF_189_) in the presence of IF and found that VEGF_165_ showed a strong ability to induce vascular sprouting and functional vessel formation when compared to VEGF_121_ or VEGF_189_.

## 2. Materials and Methods

### 2.1. Chip Design and Construction

The microfluidic chip was composed of five channels: C1, C2, C3, C4, and C5. Each channel was separated from the other by a row of posts and had an individual inlet and outlet. The width of channels C1, C2, C4, and C5 was designed to be 800 μm, and the width of channel C3 was 500 μm. The length of the interconnected section of all channels was 5.5 mm. Channel C1/C2 and C4/C5 partitioned each other with 200 μm wide pentagonal microposts spaced every 90 μm. Channel C3 is partitioned with 200 μm wide hexagonal microposts spaced every 90 μm.

A polydimethylsiloxane (PDMS) layer of the microfluidic chip was constructed by replicate molding on a master, which was coated with a 120 μm thick SU8-3035 negative photoresist (17020067; Microchem Corp, Newton, MA, USA) and patterned by photolithography. Sylgard 184 PDMS base and curing agent (Sylgard Silicone elastomer 184; Dow Corning Corp; Beijing, China) were mixed thoroughly (10:1 by volume) and poured onto the master and then put into an oven at 80 °C for 30 min. Then, the PDMS layer was peeled off the master and trimmed to size. Inlets 2/4 and outlets 2/4 were punched using a 4 mm puncher. Inlets 1/3/5 and outlets 1/3/5 were punched using a 2 mm puncher. The PDMS membrane and a confocal dish (150682; Thermofisher, Waltham, MA, USA) were treated with oxygen plasma for 1 min and then bonded together. The device was put into an oven at 80 °C for 24 h and then sterilized by ultraviolet irradiation overnight.

### 2.2. Cell Culture

Primary human umbilical vein endothelial cells (pHUVEC) were isolated from neonatal umbilical cords obtained from the Maternal and Child Health Hospital, Dalian Medical University. An umbilical cord was washed with PBS and 1% penicillin/streptomycin until the effluent buffer was transparent. Collagenase (2.5 mg/mL) solution was injected into the vein at one extremity, and the other extremity was tightly clamped with a surgical clamp. The cord was incubated in a water bath for 20–30 min. Then, the cells were collected in a centrifuge tube by washing the vein with PBS and centrifuged at 500× *g* for 10 min. The supernatant was discarded, and the pellet of cells was suspended and cultured in Endothelial Cell Medium (EnCM; ScienCell; Carlsbad, CA, USA) supplemented with 5% fetal bovine serum (FBS; Gibco, Thermofisher, Waltham, MA, USA), 1% endothelial cell growth supplement (ScienCell; USA), and 1% penicillin/streptomycin (Biological Industries; Cromwell, CT, USA). Passages from three to eight were used for angiogenic experiments. Normal human fibroblasts (NFs) were isolated from normal gingival tissues from a healthy adult during tooth extraction in the Stomatological Hospital, Dalian Medical University. Briefly, tissues were digested with collagenase type I (1 mg/mL; Sigma-Aldrich, St. Louis, MO, USA) at 37 °C with agitation for 10 h in DMEM/F12 medium (Gibco, Grand Island, NY, USA) supplemented with 10% FBS and 1% penicillin/streptomycin. The dissociated tissues were incubated without shaking for 5 min at room temperature. Then, the stromal cell-enriched supernatant was separated into a new tube and centrifuged. The cell pellet was resuspended and cultured in DMEM/F12 medium with 10% FBS and 1% penicillin/streptomycin. All cells were cultured at 37 °C in a humidified 5% CO_2_ incubator. The use of clinical samples in this study was approved by the Ethics Committee of Dalian Medical University.

### 2.3. Interstitial Flow (IF) Generation in the Chip

Fibrinogen (F8630; Sigma-Aldrich, USA) was dissolved in 0.9% NaCl at a concentration of 10 mg/mL and stored at −20 °C until use. The stored fibrinogen was thawed and diluted with Dulbecco’s phosphate-buffered saline (D-PBS; Solarbio; Beijing, China) from 10 mg/mL to 2.5 mg/mL, filter-sterilized (0.22 μm pore) and supplemented with aprotinin (0.45 TIU/mL, A1153; Sigma-Aldrich; USA). Then, the diluted fibrinogen was mixed with thrombin (1 U/mL, T4648; Sigma-Aldrich; USA) and immediately introduced into the channels C1, C3, and C5 via inlet 1, 3, and 5, and left to gel for 10 min at room temperature. After fibrin gelling, the cell culture medium was introduced into channels C2 and C4 via inlets 2 and 4. Different volumes of medium were added into C2 and C4 to generate IF in the fibrin matrix across channel C3. IF flowing from channel C2 to channel C4 was termed upstream IF, while the IF in the reverse direction was termed downstream IF. The medium in inlets C2 and C4 was changed every 12 h to maintain the pressure gradient between C2 and C4.

### 2.4. Cell Culture and Labeling in the Chip

NFs in 2.5 mg/mL liquid fibrinogen (10^7^ cells/mL) were mixed with thrombin and immediately introduced into channel C1. Acellular fibrin (2.5 mg/mL) was introduced into C3 and C5 channels. All channels were left to gel at room temperature for 10 min. After gelling, NFs were embedded in fibrinogen. EnCM medium was added to C2 and C4 channels. Then, the chip was incubated in a 5% CO_2_ incubator overnight to allow the NFs to become adapted in fibrin and EnCM medium and the next day, pHUVEC cells were seeded in channel C4. To induce angiogenesis in the chip, VEGF was introduced into channel C2, and IF was generated in channel C3. rVEGF_121_ (PeproTech; Suzhou, China), rVEGF_165_ (PeproTech; Suzhou; China), and rVEGF_189_ (ReliaTech; Saxony, Wolfenbüttel, German) were used as angiogenic stimulators. Outlets 2/4 and inlets 2/4 were kept open during the whole experiment. After angiogenesis induction, pHUVECs were labeled with TRITC Phalloidin (1:200, YEASEN, Shanghai, China) or Green Cell-Tracker (5 μM, C7025, Invitrogen; Carlsbad, CA, USA) to highlight cellular morphology. To characterize the perfusion in newly formed vessels, TRITC-dextran (70 kDa; Sigma-Aldrich; USA) was added into channel C4. Fluorescence images of the TRITC-dextran flow across C3 were recorded using a confocal microscope (Leica, TCS SP5II; Leica, Gissen, Germany). To track the flow across channel C3, FITC-labeled resin particles of 6 μm in diameter (Sigma-Aldrich; USA) were added into channel C4. Then, the movement of these particles was recorded using a confocal microscope.

### 2.5. Immunofluorescence Staining in the Chip

Paraformaldehyde (4%) was introduced into the channels C2 and C4 via inlets 2 and 4. Cells were fixed for 15 min, permeabilized with 0.15% Triton X-100 for 15 min and blocked in goat serum for 1 h at room temperature. Primary antibody anti-CD31 (1:10, Abcam; Cambridge, UK) was added into C2 and C4 channels and incubated at 4 °C overnight. The next day, Dylight488-labeled goat anti-rabbit IgG (1:200; Abbkine; Wuhan, China) was added into channels C2 and C4 and incubated for 2 h at room temperature. Then, each channel was washed with PBS three times, and images were recorded using a confocal microscope.

### 2.6. Statistical Analyses

GraphPad Prism 7.0 (Graphpad Software Inc, San Diego, CA, USA) and Image-Pro Plus 6.0 were used for statistical analyses. Differences between experimental and control groups were compared for significance using the student’s *t*-test. A *p*-value of <0.05 was considered statistically significant. Data were expressed as the mean ± SD of at least three independent experiments.

## 3. Results

### 3.1. Chip Operation and Characterization of IF

As shown in Figure 1a, channel C1 was designated as the stromal cell channel for NF cultures, C2 as the stimulation channel containing angiogenic factors, C3 as the angiogenesis channel filled with acellular fibrin, C4 as the vessel channel for endothelial cell culture, and channel C5 as the matrix channel containing acellular fibrin. When angiogenic stimulators were introduced into channel C2, they diffused in channel C3 and induced endothelial cells in channel C4 to invade channel C3.

Different volumes of medium were added into the channels C2 and C4 via respective inlets to create a hydrostatic pressure between the channels C2 and C4 thereby IF was generated in the fibrin in channel C3. The medium in inlets 2 and 4 was changed every 12 h. As shown in Figure 1b, a hydrostatic pressure of 1 mm H_2_O was generated in channel C3 at the beginning of the experiment. TRITC-dextran (70 kDa) was used to visualize the movement of fluid in channel C3. The TRITC-dextran flew from channel C2, across C3 and into C4 after approximately 480 s. The velocity of IF was approximately 1 μm/second when the hydrostatic pressure was 1 mm H_2_O. The static condition of the fluid in the fibrin in channel C3 was created by adding the same volumes of the medium into the channels C2 and C4 (Figure 1c). The TRITC-dextran remained in channel C2 and did not flow significantly into the fibrin in channel C3 under the static condition.

### 3.2. Upstream IF Enhanced Angiogenesis Induced by rVEGF_165_

To assess angiogenesis on the chip, NFs, fibrin, pHUVEC, and medium were loaded in the respective channels (Figure 2a). NFs grew in 3D fibrin and extended long processes. pHUVEC was positive for CD31 in channel C4.

The effect on angiogenesis of three experimental conditions was evaluated, including static, upstream IF, and downstream IF. After induction by rVEGF_165_ (50 ng/mL) for 72 h, pHUVEC cells grew several short sprouts under the static condition, extended numerous long vascular sprouts under the upstream IF condition, and formed a few short sprouts under the downstream IF condition (Figure 2b).

Quantitative analyses revealed that the sprout length and number was highest in the upstream IF group and lowest in the downstream IF group (Figure 2c). Analysis of sprout alignment showed that the sprouts under the upstream IF condition exhibited a clear directional preference and persistency along the streamlines of the flow (Figure 2d). Quantitative analyses revealed that the sprout numbers between −30° and 30° in the static and upstream IF groups showed significantly higher than in the downstream IF group. These findings suggested that upstream IF enhanced directional vessel sprouting induced by VEGF_165_.

### 3.3. Synergistic Effects of Upstream IF and rVEGF_165_ on Angiogenesis

We evaluated the optimal concentration of rVEGF_165_ and stimulation time for angiogenesis to occur on this chip. Different concentrations of rVEGF_165_ (0−100 ng/mL) were added into stimulation channel C2, and upstream IF was generated across C3. As shown in Figure 3a, pHUVEC did not sprout without rVEGF_165_ stimulation even in the presence of upstream IF, suggesting that VEGF played a key role in the initiation of angiogenesis when compared to IF. pHUVECs started sprouting into the fibrin in channel C3 after stimulating with rVEGF_165_ and upstream IF for 24 h. Vascular sprouts elongated with time in channel C3 toward channel C2. After 72 h, long vascular sprouts could be observed in the rVEGF_165_ groups. Tip cells with filopodia were located at the front of these sprouts. Stalk cells followed tip cells and formed vessel-like structures, which were positive for CD31. Vascular sprout invasion into the matrix was promoted progressively as the concentration of rVEGF_165_ increased. 

Quantitative analyses demonstrated that the length and number of vascular sprouts were higher in the three rVEGF_165_ groups than in the control group, and rVEGF_165_ showed the best induction effect at a concentration of 50 ng/mL (Figure 2b). Analysis of sprout alignment showed that the sprouts induced by rVEGF_165_ (50 ng/mL) exhibited a directional extension with the axis traversing the channel C3 under upstream IF condition (Figure 2c).

### 3.4. Morphology of Vascular Sprouts Induced by VEGF Isoforms 

Next, we determined sprout morphology induced by VEGF isoforms in the presence of upstream IF on the chip. We found that polarized vascular sprouts were induced by both rVEGF_121_ and rVEGF_165_. These sprouts formed CD31-positive vessel-like structures, which were hollow and lined by endothelial cells. rVEGF_165_ caused increased numbers and longer sprouts when compared to rVEGF_121_, and vessel loops could be observed between these sprouts. In contrast, the vascular sprouts induced by rVEGF_189_ were few, short, and had undetermined polarization (Figure 4a). 

Quantitative analyses revealed that the sprout length and number increased significantly in the rVEGF_165_ and rVEGF_121_ groups when compared to the rVEGF_189_ group and was highest in the rVEGF_165_ group (Figure 4b). Directional elongation of vascular sprouts could be induced by both rVEGF_121_ and rVEGF_165_ when compared to rVEGF_189_ (Figure 4c).

### 3.5. Functional Vessels Induced by rVEGF_165_


Perfusion and connectivity of vessels were demonstrated by stimulating pHUVEC with VEGF isoforms and measuring TRITC-dextran and nanoparticles on the chip. Under the stimulation of rVEGF_165_ (50 ng/mL), pHUVEC sprouts grew in the fibrin in channel C3 towards C2 with time. The sprouts elongated, the lumen diameter enlarged, and the number of vessel loops increased. By day seven, these vascular sprouts traversed channel C3 and extended from channel C4 into C2 (Figure 5a). Confocal images showed that vessel-like structures generated in C3 connected C4 with C2. These structures were hollow and lined by pHUVECs with vessel loops (Figure 5b, left). TRITC-dextran (70 kDa) was introduced into channel C4 to assess whether fluidic connections between channels C4 and C2 was established (Appendix A). It was observed that TRITC-dextran flew along these vessel-like structures and entered channel C2 without leakage into the perivascular space (Figure 5b, middle). The cross-sectional images of vessel-like structures showed that TRITC-dextran was enclosed in vessels by pHUVEC (Figure 5b, middle), indicating that pHUVEC cells formed continuous perfusion vessels with the endothelial barrier. To track the flow in newly formed vessels, solutions containing fluorescent resin particles were added into channel C4, and higher hydrostatic pressure was found to be generated in this channel than C2 (Appendix A). As visualized under the confocal microscope, some particles flew in the vessels and reached channel C2, and some particles stayed in the narrow parts of the vessels without infiltrating the perivascular matrix (Figure 5b, right). By contrast, vessels induced by rVEGF_121_ and rVEGF_189_ could not reach channel C2 after 7 days (Figure 5c). Taken together, our findings suggest that rVEGF_165_ could induce pHUVEC to generate functional vessels in the presence of upstream IF.

## 4. Discussion

In this work, we utilized a microfluidic chip to study angiogenesis, including both vascular sprouting and functional vessel formation and the key role of IF in angiogenesis was demonstrated. We further revealed the different effects of VEGF isoforms on angiogenesis. VEGF_165_ was superior to VEGF_121_ and VEGF_189_ for the induction of vascular sprouting and functional vascular formation.

In this study, we showed all the details needed to induce vessels with perfusion and connectivity functions using a microfluidic chip. Primary HUVEC cells of low passage number (<10) showed a greater propensity to form functional vessels than the HUVEC cell line. The commonly used HUVEC cell line could generate vascular sprouts on the microfluidic model but could not form functional vessels (data not shown). IF played an important role in angiogenesis [32,33,34], but it could not even stimulate endothelial cell sprouting in the absence of VEGF in this study, whereas VEGF could stimulate vessel sprouting in the absence of IF. This suggested that IF was not an initiator of vascular sprouting. However, the direction of vascular sprouting was greatly influenced by IF [35,36]. In our study, vascular sprouts were not polarized without IF, and their direction was consistent with the direction of IF. It is also notable that vascular sprouts induced by VEGF could not mature into functional vessels without IF (data not shown). 

VEGF isoforms are generally co-expressed in all tissues, with VEGF_165_ being the predominant form [37,38]. The different isoforms show similar functions as potent stimulators of angiogenesis but differ in their binding affinity for the ECM [39]. VEGF_121_ is the shortest isoform and is secreted into the extracellular space as a soluble peptide. However, VEGF_165_ can work as a soluble form or is bound to heparan sulfate proteoglycans (HSPG) in the ECM. VEGF_189_ has the highest affinity for the ECM by binding to HSPG [40,41]. Limited information is available on the extent of induction of angiogenesis by the different VEGF isoforms. In this study, we compared the characteristics of angiogenesis induced by three human VEGF isoforms. We demonstrated that VEGF_165_ was the most potent at inducing vascular sprouting in vitro and could also induce functional vessel formation. A different study has compared the characteristics of angiogenesis in tumors induced by mouse VEGF isoforms. The mouse VEGF proteins are one residue shorter than human VEGF and thus produce VEGF_120_, VEGF_164_, and VEGF_188_. The work from this study established xenograft models using isoform-specific expressing tumor cells and demonstrated that mouse VEGF_120_ recruited peritumoral vessels but hardly vascularized the tumor itself, whereas VEGF_164_ could both recruit vessels and vascularized the tumor, and VEGF_188_ recruited the host vasculature weakly [42]. It is clear that further study is required to elucidate the exact mechanisms by which VEGF isoforms induce different morphologies of angiogenic sprouts.

## 5. Conclusions

In this study, we utilized a microfluidic chip for the induction of angiogenesis, and this platform enabled the analysis of angiogenesis stimulated by both angiogenic factors and IF. Both angiogenic sprouting and functional vessel formation could be reproduced on this chip. We found that VEGF was a determinant of the initiation of vascular sprouts, and the direction of vascular sprouts was greatly influenced by IF. VEGF isoforms displayed different angiogenic potentials on the chip as assessed by the sprout length and number and vessel perfusion and connectivity. Taken together, we believe that this chip is applicable to analyze the effects of angiogenic factors or drugs on angiogenesis, as well as to explore the mechanisms of angiogenesis induced by VEGF isoforms.

## Figures and Tables

**Figure 1 micromachines-13-00225-f001:**
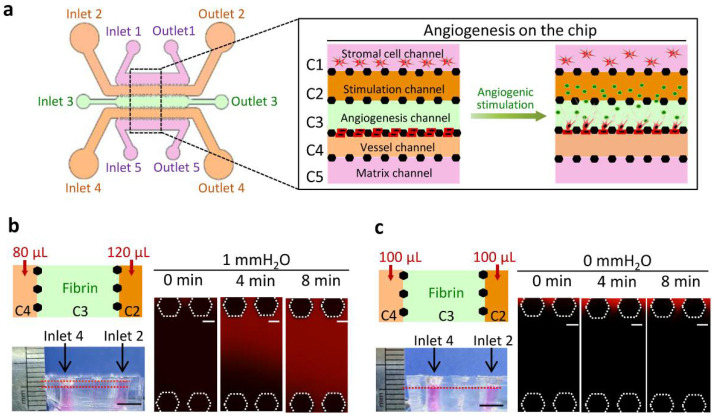
Chip design and characterization of IF. (**a**) Schematic overview (left) and channel configuration (right) of the microfluidic chip designed for the induction of angiogenesis. (**b**) IF generation in fibrin in channel C3. Left top: Different volumes of media were added into the channels C2 (120 μL) and C4 (80 μL) to produce a hydrostatic pressure difference, and they were changed every 12 h. Hydrostatic pressure of 1 mmH_2_O was generated when the media was just added into the channels C2 and C4. Left bottom: A Photo showed the sideview of inlet volumes. Right: Fluorescence images of TRITC-dextran (70 kDa, red) flowing through acellular fibrin in the channel C3 at 0, 240, and 480 s under the hydrostatic pressures of 1 mmH_2_O. (**c**) Static condition in the fibrin in channel C3. Left top: The same volumes of media were added into the channels C2 and C4 (100 μL), and the media were changed every 12 h. Left bottom: A photo showed the sideview of inlet volumes. Right: Fluorescence images of TRITC-dextran (70 kDa, red) in fibrin at 0, 240, and 480 s under the hydrostatic pressures of 0 mmH_2_O. Scale bars, 5 mm (photos in (**b**,**c**)) and 100 μm (fluorescence images in (**b**,**c**)).

**Figure 2 micromachines-13-00225-f002:**
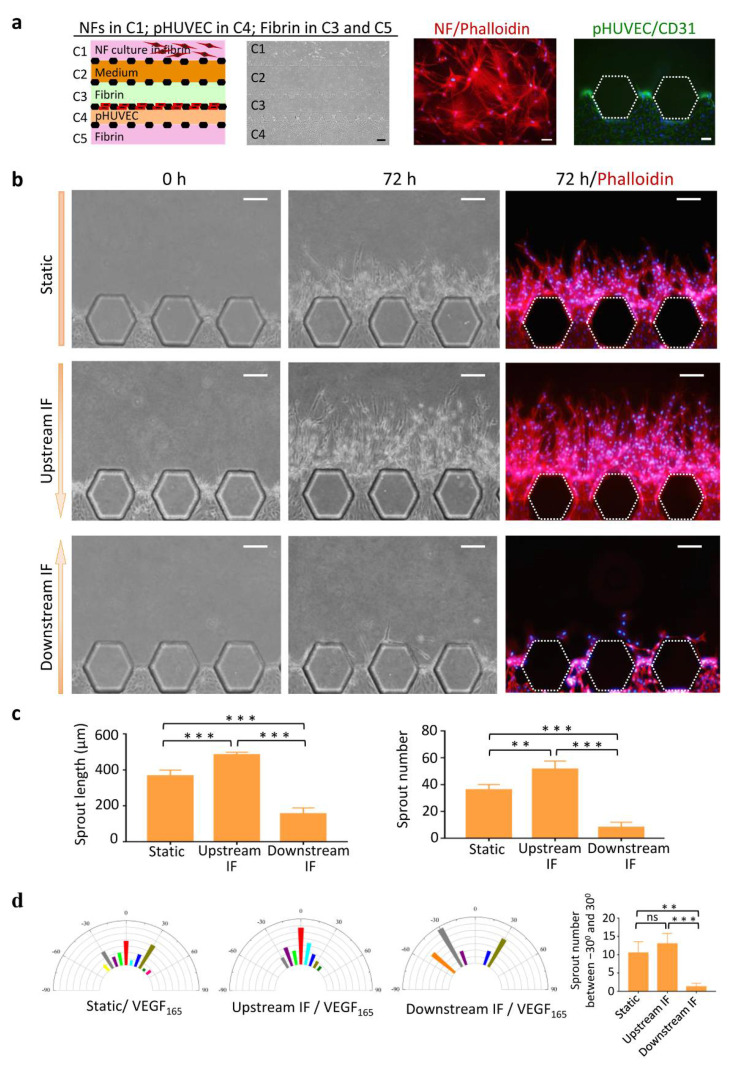
Effects of IF on angiogenesis induced by rVEGF_165_. (**a**) Cell seeding and fibrin loading in the chip. Illustration and bright field of cells and fibrin in individual channels (left). Scale bar = 200 μm. Morphology of NFs in C1 stained with phalloidin (middle) and pHUVEC in C4 stained with CD31 (right). Scale bar = 50 μm. (**b**) Images of vascular sprouts induced by rVEGF_165_ (50 ng/mL) under the static (up panel), upstream IF (middle panel), and downstream (low panel) conditions. (**c**) Quantitative analyses of angiogenesis in terms of average sprout length and number. (**d**) Semicircular histograms showing the orientation angle distribution of the vascular sprouts in static (left), upstream IF (middle left), and downstream (middle right) conditions. Each Bar with different colors in the histograms represents the percentage of sprout orientation across the population of sprouts grown under the indicated condition. Orientation 0 represents the alignment of the sprout with the axis traversing channel C3. Quantitative analyses of sprout orientation (right). Scale bar = 100 μm. ** *p* < 0.01, *** *p* < 0.001.

**Figure 3 micromachines-13-00225-f003:**
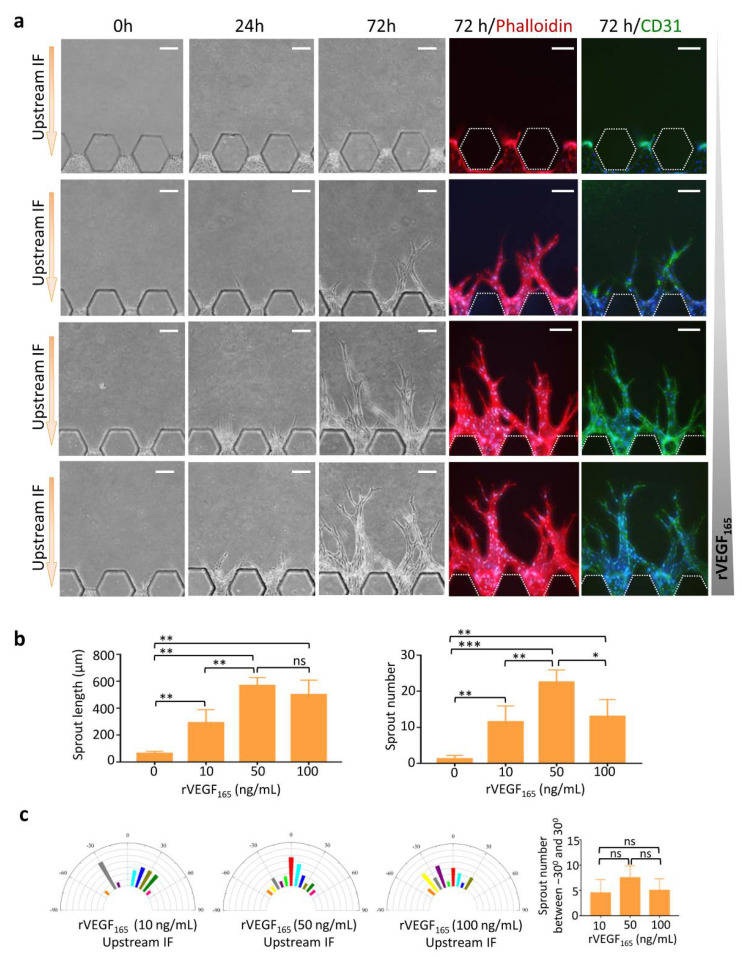
Optimal concentration of rVEGF_165_ for angiogenesis. (**a**) Images of vascular sprouts induced by rVEGF_165_ (0−100 ng/mL) in the presence of upstream IF. (**b**) Quantitative analyses of angiogenesis in terms of average sprout length and number. (**c**) Semicircular histograms showing distribution of orientation angle of the vascular sprouts stimulated by 10 ng/mL (left), 50 ng/mL (middle left), and 100 ng/mL (middle right) rVEGF_165_. Orientation of 0 represents the alignment of the sprout with the axis traversing channel C3. Quantitative analyses of sprout orientation (right). Scale bar = 100 μm. * *p* < 0.05, ** *p* < 0.01, *** *p* < 0.001.

**Figure 4 micromachines-13-00225-f004:**
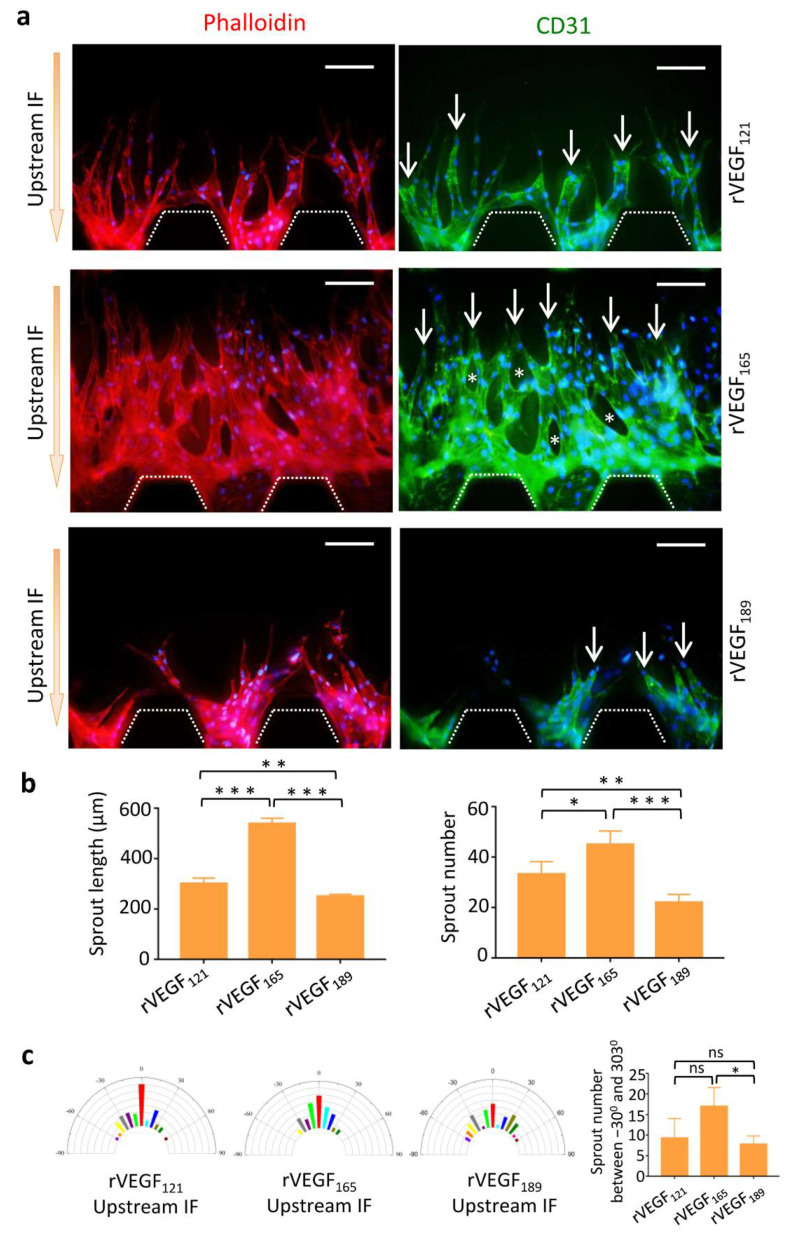
Morphologies of vascular sprouts induced by rVEGF_121_, rVEGF_165_, and rVEGF_189_. (**a**) Images of vascular sprouts induced by rVEGF_121_, rVEGF_165_, rVEGF_189_ in the presence of upstream IF. rVEGF_165_ induced increased numbers and longer sprouts (arrows) compared to rVEGF_121_ and VEGF_189_. Vessel loops (*) could be observed between the sprouts in VEGF_165_. (**b**) Quantitative analyses of angiogenesis in terms of average sprout length and number. (**c**) Semicircular histograms showing the distribution of orientation angle of vascular sprouts stimulated by rVEGF_121_ (left), rVEGF_165_ (middle left), and rVEGF_189_ (middle right) with the upstream IF. Quantitative analyses of sprout orientation (right). Scale bar = 100 μm. * *p* < 0.05, ** *p* < 0.01, *** *p* < 0.001.

**Figure 5 micromachines-13-00225-f005:**
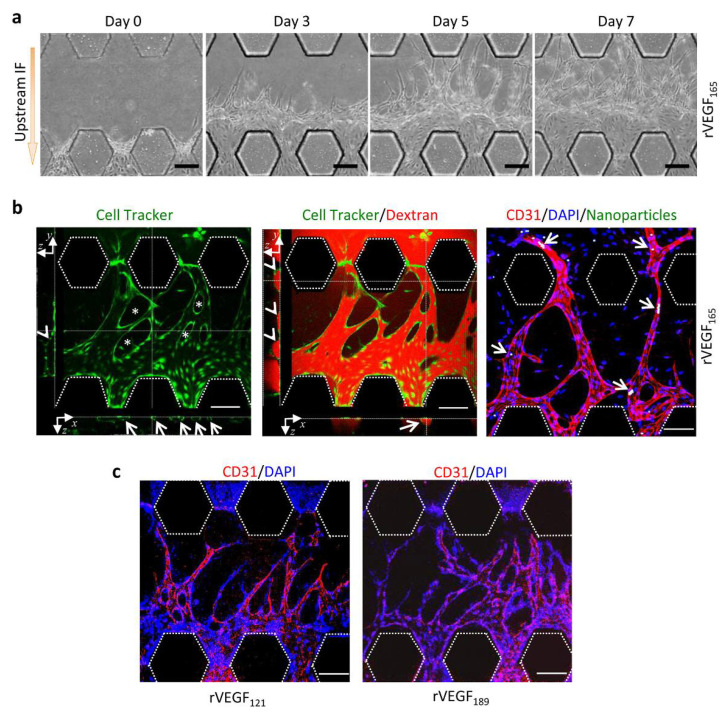
Functional vessels induced by rVEGF_121_, rVEGF_165_, and rVEGF_189_. (**a**) Images of vascular sprouts induced by rVEGF_165_ in the presence of upstream IF with time. (**b**) Vessel morphology. Cells were stained with CellTracker (green). Several vessel loops (*) could be observed. X−z cross-sectional image showed that five vessels were hollow and lined by a layer of cells (arrows). Y−z cross-sectional image showed that a hollow vessel was lined by a layer of cells (arrowhead) (left). Dextran permeability assay. X−z (arrow) and y−z (arrowheads) cross-sectional images showed that TRITC-dextran (red) was enclosed in vessels by pHUVEC (green) (middle). Nanoparticle tracking assay (right): Nanoparticles (arrows, white) entrapped in newly formed vessels, which were positive for CD31. (**c**) CD31-positive vessels induced by rVEGF_121_ and rVEGF_189_. Scale bar = 100 μm.

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
