# Peer review of "Angiogenesis and Functional Vessel Formation Induced by Interstitial Flow and Vascular Endothelial Growth Factor Using a Microfluidic Chip"

_micromachines, 2022, doi:10.3390/mi13020225_

Round 1

Reviewer 2 Report

Comments to the Author:
In this manuscript, the authors demonstrated an approach for reconstructing an angiogenic model using a microfluidic chip and investigated the morphology of angiogenic sprouting and vessel functions. This chip is applicable to study the influence of angiogenic factors or dugs on angiogenesis. However, I have several concerns before this manuscript can be accepted. Therefore, in its current form, revisions are needed.

1.What is the dimensions of this chip, especially the sizes of the different channels? These should be given in the manuscript.

2. What is the flow rate of different reagents used in this experiment? 

3. Will the chip structure or the sample flow rate influence the cells’viability, the Angiogenesis and functional vessel formation.

Reviewer 3 Report

Angiogenesis and functional vessel formation induced by interstitial flow and vascular endothelial growth factor using a microfluidic chip 

[Figure 1 b.d] Please include a more accurate scale bar that is not part of a ruler. Also please indicate how long is the error bar in Figures c and e.

[line 171, major] Please specify your  IF. for 72 hours, did you change media? Was it a continuous flow over the 72 hours? what is the flow speed profile variation with respect to time? It is well documented that sprouting is affected not only the flow but also the change of the flow. 

[Line 190, major] please indicate the statistical method used, is it a T-test or is it ANOVA. Please also indicate what *** means for the P-value. 

[Figure 3.c] There is a visual difference in the directional spread across the three conditions, would be good to include a statistical study on it. 

[Figure 4] See comment [Line 190, major] 

[Figure 5] Please indicate what is the difference made by 121, 165 vs 189, and please include statistical evidence. This part of the study needs to be properly motivated and concluded as to why you do it and what did you found. 

Reviewer 4 Report

The authors presented an interesting work about the in vitro angiogenesis study, using a microfluidic chip, an experimental platform able to mimic the microenvironment in vivo as closely as possible.

  • The manuscript is well written, it just needs a careful form revision to fix some oversights.
  • As indicated in the guidelines for authors, all words in titles should be capitalized, except conjunctions, articles, and all prepositions. Check the various points of the guidelines carefully.
  • Concerning the experimental section, all procedures and results are clearly and completely described. One suggestion: in section 2 (2.2 cell culture) the authors reported the isolation of primary human umbilical vein endothelial cells (pHUVEC) and normal human fibroblasts (NF), without the protocol description in detail. Considering that it is part of the materials and methods section, this procedure (extraction protocol used) must also be added.

Round 2

Reviewer 1 Report

good job, thank you